# Adding Concomitant Chemotherapy to Postoperative Radiotherapy in Oral Cavity Carcinoma with Minor Risk Factors: Systematic Review of the Literature and Meta-Analysis

**DOI:** 10.3390/cancers14153704

**Published:** 2022-07-29

**Authors:** Alessia Di Rito, Francesco Fiorica, Roberta Carbonara, Francesca Di Pressa, Federica Bertolini, Francesco Mannavola, Frank Lohr, Angela Sardaro, Elisa D’Angelo

**Affiliations:** 1Radiotherapy Unit, P.O. “Mons. A.R. Dimiccoli”, Viale Ippocrate 15, 76121 Barletta, BT, Italy; alessia.dirito@aslbat.it; 2Department of Radiation Oncology and Nuclear Medicine, State Hospital Mater Salutis AULSS 9, Via Carlo Gianella, 1, 37045 Legnago, VR, Italy; francesco.fiorica@aulss9.veneto.it; 3Radiation Oncology Department, General Regional Hospital “F. Miulli”, 70021 Acquaviva delle Fonti, BA, Italy; 4Radiotherapy Unit, Azienda Ospedaliero Universitaria di Modena, Via del Pozzo 71, 41121 Modena, MO, Italy; 297014@studenti.unimore.it (F.D.P.); frank.lohr@unimore.it (F.L.); dangelo.elisa@aou.mo.it (E.D.); 5Division of Medical Oncology, Azienda Ospedaliero Universitaria di Modena, Via del Pozzo 71, 41121 Modena, MO, Italy; bertolini.federica@policlinico.mo.it; 6Division of Medical Oncology, AOU Consorziale Policlinico di Bari, 70124 Bari, BA, Italy; f.mannavola1@studenti.uniba.it; 7Interdisciplinary Department of Medicine, Section of Radiology and Radiation Oncology, University of Bari “Aldo Moro”, 70124 Bari, BA, Italy; angela.sardaro@uniba.it

**Keywords:** postoperative radiochemotherapy, adjuvant chemoradiotherapy, oral cavity cancers, minor risk factors, intermediate risk factors

## Abstract

**Simple Summary:**

Oral cavity carcinoma (OCC) is the 11th most frequently diagnosed cancer; despite a multimodal treatment, locally advanced OCC, managed by surgery and adjuvant therapies, remains at high risk of recurrence, with a 5-year overall survival (OS) of 51%. The efficacy of postoperative chemotherapy in addition to radiotherapy (POCRT) in low–intermediate risk OCC is a controversial matter in the absence of high-risk features (ENE, R1). To establish the role of POCRT in a population with solely minor risk factors (perineural invasion or lymph vascular invasion; pN1 single; DOI ≥ 5 mm; close margin; node-positive level IV or V; pT3 or pT4; multiple lymph nodes without ENE), we performed a systematic review and meta-analyses focused on OS, disease-free survival (DFS), and local-recurrence-free survival (LRFS). Thirteen studies met the inclusion criteria and were included in the quantitative meta-analyses. Our preliminary results are in favor of POCRT in terms of OS but not conclusive for DFS and LRFS. Further analyses are suggested.

**Abstract:**

When presenting with major pathological risk factors, adjuvant radio-chemotherapy for oral cavity cancers (OCC) is recommended, but the addition of chemotherapy to radiotherapy (POCRT) when only minor pathological risk factors are present is controversial. A systematic review following the PICO-PRISMA methodology (PROSPERO registration ID: CRD42021267498) was conducted using the PubMed, Embase, and Cochrane libraries. Studies assessing outcomes of POCRT in patients with solely minor risk factors (perineural invasion or lymph vascular invasion; pN1 single; DOI ≥ 5 mm; close margin < 2–5 mm; node-positive level IV or V; pT3 or pT4; multiple lymph nodes without ENE) were evaluated. A meta-analysis technique with a single-arm study was performed. Radiotherapy was combined with chemotherapy in all studies. One study only included patients treated with POCRT. In the other 12 studies, patients were treated with only PORT (12,883 patients) and with POCRT (10,663 patients). Among the patients treated with POCRT, the pooled 3 year OS rate was 72.9% (95%CI: 65.5–79.2%); the pooled 3 year DFS was 70.9% (95%CI: 48.8–86.2%); and the pooled LRFS was 69.8% (95%CI: 46.1–86.1%). Results are in favor of POCRT in terms of OS but not significant for DFS and LRFS, probably due to the heterogeneity of the included studies and a combination of different prognostic factors.

## 1. Introduction

Oral cavity carcinoma (OCC) is listed as the 11th most frequently diagnosed cancer (2.0% of all cancers) [1] and as the 15th most frequently deadly cancer, characterized by a 5-year overall survival of 51%.

Despite a multimodal treatment, locally advanced OCC, managed by surgery and adjuvant therapies, remains a disease with high risk of recurrence [2]. Classically, postoperative chemoradiotherapy (POCRT) is recommended for patients with major pathological risk factors (RFs); i.e., positive surgical margins (R1) and/or lymph node metastases with extracapsular extension (ECE). These recommendations are based on the results of two landmark prospective trials published in 2004: the RTOG 9501 and the EORTC 229313 [3,4]. Both trials showed improved local control rate (LCR) and disease-free survival (DFS) in high-risk patients addressed with POCRT, but the selection of patients was slightly different. In the RTOG trial, patients with two or more positive regional lymph nodes, ECE of nodal disease and microscopically involved margins of resection were included. In the EORTC trial, patients with pT3/T4, pN2–N3 or pT1-2N0-1 with ECE; positive resection margins; perineural invasion (PNI); lymph vascular invasion (LVI); or oral cavity/oropharyngeal cancers with lymph node metastasis at level IV or V were included, and thus wider inclusion criteria were applied in this study. In both trials, acute toxicities were significantly increased with POCRT compared with PORT alone. Late toxic effects, and particularly swallowing dysfunction, were also common in these patients. A subsequent pooled analysis of the two studies published in 2005 by Bernier et al. [5] concluded that chemotherapy concurrent with post-operative radiotherapy was not indicated in the absence of major RFs (R1, ECE). In any case, in oral cavity carcinoma, considering the poor prognoses at the locally advanced stage, great attention has been given to the role of minor RFs and their implications for treatment approach. Suggested minor RFs include pT3–T4 disease, pN2–pN3, pathological N1 disease level IV–V, surgical margin ≤5 mm, perineural invasion, vessel invasion, and lymph vascular invasion [6], but others have recently been postulated (i.e., tumor invasion depth ≥ 11 mm, G3). However, the published results were overwhelmingly derived from retrospective studies, so their role in prognosis is still a matter of on-going debate.

Specifically, some authors have provided evidence that a combination of multiple minor RFs is associated with poor prognosis, thus suggesting a role for more intensive treatment, such as chemotherapy concurrent with PORT, in selected populations [7].

The aim of this systematic review with meta-analysis was to assess the potential role of chemotherapy added to post-operative radiotherapy in locally advanced OCC patients presenting with solely minor RFs. We analyzed the literature in terms of oncological outcomes.

## 2. Materials and Methods

A systematic review following the PRISMA methodology [8] was performed with the following research question: what is the impact of chemotherapy added to post-operative radiotherapy in terms of oncological outcomes in patients with solely minor risk factors?

This systematic review was registered with the International Prospective Register of Systematic Reviews (PROSPERO) under the number CRD42021267498. A literature search according to the population, intervention, comparison, outcome, and timing (PICOT) model (Table 1) was independently conducted by two authors (E.D., A.D.R.) to identify articles published in the PubMed, EMBASE, and Cochrane databases between January 2000 and 23 November 2021.

A combination of the terms “oral cancer”, “post-operative radiotherapy”, “POCRT”, and “adjuvant chemoradiotherapy” were used. The search strings for each database are depicted in the Appendix A. We began our search from 2000 because the landmark trials mentioned above were published in 2004. After removal of duplicates, we analyzed the remaining articles for selection (using the title/abstract) and then for eligibility (using the full text) following the PRISMA statement (Figure 1).

### 2.1. Inclusion and Exclusion Criteria

Studies were included only if they analyzed outcomes of patients with oral cancer treated with radiotherapy or chemoradiotherapy who had minor pathological risk factors: perineural invasion or lymph vascular invasion; pN1 single; DOI ≥ 5 mm; close margin (<2–5 mm); node-positive level IV or V; pT3 or pT4; multiple lymph nodes in the absence of ENE. At least 40% of the studied population had to have oral cancer. Abstracts, letters, proceedings from scientific meetings, editorials, expert opinions, reviews without original data, case reports, studies using proton-ion carbon therapy and brachytherapy, repetitive data, non-English language papers, phase I studies, translational studies, and animal studies were excluded. Studies where major and minor risk factors were studied together were also excluded.

In some cases, the full texts of the retrieved papers were also analyzed to identify additional references satisfying the inclusion criteria. Studies included were prospective or retrospective, analyzing more than 10 patients. Only studies including oncological outcomes (overall survival, disease-free survival, and local recurrence-free survival) for patients affected by oral cavity carcinoma (OCC) treated with post-operative chemoradiotherapy who showed the presence of minor risk factors were analyzed. Criteria for exclusion are reported in the Appendix A [9,10,11,12,13,14,15,16,17,18,19,20,21,22,23,24,25,26,27,28,29,30,31,32,33,34,35,36,37,38,39,40,41,42,43,44,45,46,47,48,49,50,51,52,53,54,55,56,57,58,59,60,61,62,63,64,65,66,67,68,69,70,71,72]. Discrepancies in study selection were infrequent (overall inter-observer variation < 10%) and were solved through discussion between the four authors (E.D., A.D.R., R.C., F.F.).

### 2.2. Review of the Trials

After study selection, the methodological quality of the studies was assessed according to a checklist for quality appraisal of case series studies produced by the Institute of Health Economics (IHE) [73].

A meta-analysis technique with a single-arm study was used to determine the pooled 3-year overall survival, 3 year disease-free survival, and 3 year local recurrence-free survival. We calculated the estimated population proportions for 3-year overall survival, disease-free survival, and local recurrence-free survival with 95%CIs for each separate study. Pooled effect size aided the general evaluation of the treatment effect. Heterogeneity across studies was examined using the Cochran *Q* chi-squared test and I^2^ statistics. Studies with I^2^ statistics of 25–50%, 50–75%, and >75% were deemed to have low, moderate, and high heterogeneity, respectively [74]. We used random effects models because there was great subjectivity, given the lack of related control groups, in the non-comparative studies and a tendency toward high heterogeneity.

## 3. Results

The search of the literature yielded 11,041 citations. After removal of duplicates, a total of 9125 citations remained, and 7899 of those were excluded using the title and abstract (respectively, 7996 and 1048), leading to 82 studies being assessed for relevance using a full-text review (the main reasons for exclusion are described in Appendix A). Of these, 12 studies met the inclusion criteria. Decisions on which studies to include were made blindly by three reviewers (E.D.A, A.D.R, and F.F.). Disagreements were resolved by discussion. Studies were excluded if they were: review articles, guidelines, commentaries, editorials, letters not in English, ex vivo or in vivo studies, or studies describing patients with major and minor risk factors without discriminating between different results for the population.

The main features of the trials included in this systematic review are shown in Table 2. These studies were published between 2010 and 2021 in three countries. Only one study was a prospective trial [75]; the others were retrospective. The analyzed population in each study varied greatly, ranging from 425 to 10,870 patients. The 12 studies included 24,569 patients treated with radiotherapy or radiochemotherapy for oral cancer with minor risk factors and with negative SM and no ECE. Treatment was delivered with adjuvant intent in all studies. Median follow-up from diagnosis was 38.4 months, ranging from 24 to 130 months. The main characteristics of the studies included in the analysis are shown in Table 2 and Table 3.

### 3.1. The Site, Dose, and Interval of Radiotherapy

Head and neck irradiation was reported for local advanced oral cancer after surgery with no major risk. The median radiation dose was 72.5 Gy in all studies; there was no information about the fractionation used in most studies. The information about radiation techniques was presented in six studies [75,77,78,79,80,83]; only two declared the exclusive use of static intensity-modulated radiotherapy (IMRT) or volumetric modulated arc therapy (VMAT) [78,79]. Radiotherapy was combined with chemotherapy in all studies. One study [77] only included patients treated with radiochemotherapy. In the other eleven studies, patients were treated with only adjuvant radiotherapy (12,883 patients) or with POCRT (10,663 patients).

### 3.2. Survival Analysis

All studies but one [80] reported the 3-year OS rate. Analyzing the use of adjuvant treatment (radiotherapy with or without chemotherapy), globally, the pooled 3-year OS rate was 66% (95%CI: 57.2–73.8%) with high heterogeneity (I^2^ = 98.7%, *p* < 0.0001) (Figure 2a).

Five studies [77,78,79,81,82] reported the 3-year disease-free survival. Globally, the pooled 3-year disease-free survival rate was 63.7% (95%CI: 54.7–71.9%) with moderate heterogeneity (I^2^ = 50.9%, *p* = 0.08) (Figure 2b). Only four studies [78,79,82,83] evaluated 3-year local recurrence-free survival. The pooled analysis showed that 75.4% (95%CI: 61.1–86.7%) of patients experienced a loco-regional control at three years with low heterogeneity (I^2^ = 84.1%, *p* = 0.0003) (Figure 2c).

Among all patients, 12,883 were treated with radiotherapy across nine studies [76,77,78,79,81,82,84,85,87], and the pooled 3 y OS rate was 61.2% (95%CI: 50.7–70.7%) with high heterogeneity (I^2^ = 99.2%, *p* < 0.0001) (Figure 3a). Three studies [78,79,82] reported disease-free survival. The pooled 3-year disease-free survival was 62% (95%CI: 44.6–76.8%) with low heterogeneity (I^2^ = 23.9%, *p* = 0.268) (Figure 3b). Only two studies [78,82] reported 3-year local recurrence-free survival; the pooled LRFS was 72.6% (95%CI: 17–97.2%) with low heterogeneity (I^2^ = 0, *p* = 0.4346) (Figure 3c).

Among all patients, 10,663 were treated with radiochemotherapy across 10 studies [76,77,78,79,81,82,83,84,85,87], and the pooled 3 y OS rate was 72.9% (95%CI: 65.5–79.2%) with high heterogeneity (I^2^ = 91.5%, *p* < 0.0001) (Figure 4a). Four studies [75,78,79,82] reported disease-free survival. The pooled 3-year disease-free survival was 70.9% (95%CI: 48.8–86.2%) with high heterogeneity (I^2^ = 74.4%, *p* = 0.008) (Figure 4b). Only three studies [75,78,82] reported 3-year local recurrence-free survival, and the pooled LRFS was 69.8 (95%CI: 46.1–86.1%) with moderate heterogeneity (I^2^ = 53.8%, *p* = 0.115) (Figure 4c).

Globally, the results of the meta-analysis for the analyzed populations in terms of OS, DFS, and LRFS are summarized in the graph in Figure 5.

## 4. Discussion

### 4.1. Analysis of Minor Risk Factors

Optimizing adjuvant therapy for patients with resected locally advanced head and neck cancers remains a persistent challenge almost two decades after initial reports from the RTOG 95-01 and EORTC 22931 trials [5]. The conflicting trial results leave clinicians with a lack of clear level 1 evidence to select patients for POCRT. The pooled analysis of these trials and long-term results from RTOG 95-01 [4] indicate that patients with ECE or positive margins are most likely to benefit from POCRT. A possible explanation of these results is that substantially fewer events occurred in patients not displaying positive margins and/or extracapsular extension features. Therefore, the two randomized clinical trials were not designed to assess the results in subgroup analysis and, in reference to OCC, it should be noted that the frequency of this disease in these investigations accounted for only 25% of the total cases. Finally, the lack of overlapping intermediate-risk factors between the trials altered the combined analysis to assess the impact of CRT in these patients. However, practice guidelines [6] suggest that either POCRT or PORT may be appropriate for patients with other minor risk factors.

This systematic review with meta-analysis aims to help clinicians understand when it would be more useful to associate concomitant chemotherapy with PORT in patients with resected OCC and only minor risk factors.

We analyze in detail each minor risk factor and the potential role of each of them in the choice of a combined postoperative treatment, with reference to the case series included in the review.

#### 4.1.1. Depth of Invasion (DOI)

With the advent of the eight edition of the AJCC staging system, the depth of invasion (DOI) replaced the previous prognostic factor of tumor thickness (TT).

The role of DOI as a prognostic factor is based on an international collaborative retrospective study: Ebrahimi [87] demonstrated that DOI, with two different cut-offs of 5 and 10 mm, was an independent prognostic factor for disease-specific survival (DSS) in a multivariate analysis of 3149 patients with OSCC. Some authors have suggested that a clear cut-off indicating further need of treatment intensification is far from being identified due to a concern about a univocal diagnostic criterion for DOI [88]. Recently, Shinn et al. [89] reported that the risk of regional recurrence in oral tongue carcinomas increased progressively with any DOI [89].

As emerged from a recent retrospective study in early-stage oral tongue patients [89], the presence of DOI alone is not sufficient to indicate the need for PORT.

In any case, a recent meta-analysis showed that DOI has a correlation with the probability of development [90] of nodal metastases in the early stage; however, its role as an isolated risk factor remains to be demonstrated [87].

#### 4.1.2. Perineural Invasion (PNI)

According to different case series, in oral cavity carcinoma, perineural invasion rates vary from 6% to almost 80%; thus, it often co-exists with other prognostic factors. In fact, establishing a clear role for the oncological outcome of isolated PNI and, conversely, the need for potential treatment intensification remains difficult [91,92,93,94,95,96,97].

It has been reported that perineural invasion is associated with a decreased rate of overall survival, increased rate of local recurrence and regional metastasis, and increased disease-specific mortality [97,98,99,100,101].

In the surgical series published by Zanoni, PNI was significantly associated in the multivariate analysis with 5-year overall (risk ratio (RR): 1.26, 95%CI: 1.05–1.51; *p* = 0.012) and disease-specific survival (RR: 1.36, 95%CI: 1.03–1.79; *p* = 0.028) [71]. The multivariate analysis of a multi-institutional collaborative group study (196 POCRT patients, 128 PNIs) [102] showed that DFS was significantly worse in those with PNI (HR: 3.08, 95%CI: 1.71–5.53; *p* < 0.001) and significantly better in those receiving at least 200 mg/m^2^ adjuvant cisplatin (HR: 0.951, 95%CI: 0.91–0.99; *p* = 0.007).

A great deal of the literature evidence shows an association of PNI with mortality and the risk of recurrence, but the different histopathological definitions of infiltration (surrounding at least 1–3 of the circumferences of the nerve or only touching it) could be the reason for the non-homogeneous evidence in the literature [103,104,105].

Despite its potential prognostic role, perineural invasion has not been incorporated in the latest version of the AJCC system [106].

#### 4.1.3. Lymph Vascular Invasion (LVI)

Many authors try to assess the prognostic role of lymph vascular invasion in OCC. Huang [107], in his meta-analysis conducted up to 2020, included 36 studies involving 17,109 patients. The results showed that the presence of LVI was associated with poor OS and DSS in OSCC and a trend toward lymph node metastasis in the early stage, suggesting a role for intensive treatment in this subset of patients. Similar results were obtained in the meta-analysis by Dolens; however, the authors underline the low evidence level for both OS and DFS because of the large heterogeneity in the reported HRs [108].

As for perineural invasion, an important issue concerns the definition of LVI. Fives [109] and Mascitti [110] defined LVI as the presence of tumor cells within the vessels; in a large proportion of studies, lymphatic or vascular invasion were classified together under the concept of LVI.

Since the potential roles of perineural invasion, lymph vascular invasion and DOI have only recently been suggested, the literature analyzed often had aggregated or missing data, or did not specify rates, for these risk factors. In any case, we here try to summarize the main evidence.

In the study by Chen WC [78], the presence of minor risk factors, such as perineural, lymphatic, or vascular invasion, was low, accounting for 10.9%, 2.6%, and 1.6% of patients, respectively. Conversely, tumor depth (< 10 mm) was present in 55.7%. In any case, these authors found that, in the multivariate analysis with patients with a combination of at least three minor risk factors (179 patients), POCRT yielded significantly better 5-year LRC, DFS, and OS compared to surgery only or the PORT group. In the NCDB analysis (2803 patients with oral tongue cancer treated from 2004 to 2012) conducted by Spiotto and colleagues [76] with the aim of assessing the role of POCRT in patients affected by OSCC in the presence of intermediate (minor) risk factors, the multivariate analysis did not highlight an improvement in OS for patient tumor depth >5 mm and LVSI, and no data were available about perineural invasion. In a similar study using the NCDB conducted by Trifiletti [77], data on lymph vascular invasion were collected but not analyzed because a significant portion were lacking (68%). In the retrospective analysis by Fan [13] of patients with at least three minor risk factors, the multivariate analysis indicated that a tumor invasion depth ≥ 11 mm was an independent poor prognostic factor (*p* < 0.05). In contrast, Tasoulas [86], in a mixed population of HNSCC patients (42 patients with OCC, 23% of total, intermediate-risk patients), found that the addition of chemotherapy to RT increased mortality risk. In the study by Chen MM [81], a wide analysis of the NCDB for HNSCC patients subjected to PORT or POCRT—of whom 1571 (62.2%) were treated with pRT and 956 (37.8%) with pCRT, respectively, with the presence of multiple positive lymph nodes and lymph vascular invasion in about 50% of patients in both groups—did not find differences in mortality with or without lymph vascular invasion. In any case, due to concerns regarding multiple sub-group analyses and the small number of patients who received chemotherapy, LVI was not analyzed as an independent risk factor. In the retrospective evaluation of the National Cancer Registry by Fan [82] of a selected population of patients with multiple nodal metastases associated only with other minor risk factors, despite an imbalance in the characteristics of the POCRT arm (perineural and pT stage, *p* < 0.05), POCRT was correlated at multivariate analysis with better treatment outcomes.

#### 4.1.4. pN1

Among patients with a single positive minor lymph node of 3 cm without ECE, the benefit of adjuvant treatment is not clear. According to NCCN guidelines [6], RT can be considered for patients with N1 disease and no adverse features in the oral cavity or supraglottic larynx subsites. This type of patient is rarely included in the RTOG and EORTC landmark studies; therefore, it is uncertain from the combined analysis whether chemoradiation is beneficial for this setting. In the retrospective study by Lee et al. [37], nearly a third of the patients who underwent adjuvant therapy received trimodal therapy with concurrent chemotherapy, which is even more surprising in this setting of unclear efficacy. One potential reason was the presence of lymph vascular invasion and/or perineural invasion, each of which was included in the EORTC 22931. Trifiletti et al. [77], examining locally advanced (stage III–IV) head and neck cancers that were treated with PORT or POCRT, selected patients with negative surgical margins and no ECE. After propensity matching, the authors still found a small but statistically significant improvement in OS among patients with a single positive node, probably due to the presence of other minor risk factors at the same time. From these experiences we could argue that pN1 alone does not represent a risk factor indicating the need for the addition of concomitant CT to PORT, but, together with other minor risk factors, such as PNI and LVI, it must be considered for treatment intensification.

#### 4.1.5. Multiple Positive Nodes

As in other common malignancies, such as breast cancer, in HN cancers the number of positive nodes is related to a worse prognosis. For OCC, several papers [111,112] have demonstrated that the number of metastatic lymph nodes (LNs) is the most important factor influencing survival, exceeding the impact of classic high-risk factors, such as ENE and positive margins. Many of the studies excluded from our review due to the abstract or full text had prescribed POCRT not only in patients with positive margins and/or ECE, but also in cases of pN2, reflecting the use of POCRT in different real-world clinical settings. Therefore, excluding patients with multiple positive nodes from POCRT is worrisome.

Zumsteg [113] analyzed 7144 HNC patients from the NCDB database (68.4% OSCC) and showed that increasing metastatic nodal burden was associated with increased impact on survival from POCRT vs. PORT. The authors compared their results with the combined analysis of EORTC/RTOG, and they hypothesized that many patients with positive margins or ENE in the combined analysis probably also had high metastatic nodal burden, making it difficult to separate the independent effects of these correlated variables. Moreover, they excluded oropharyngeal cancers (representing 30% and 42% of the EORTC and RTOG cohorts, respectively), which are biologically and clinically distinct from other head and neck cancers [114] and may have a different relationship with nodal burden, ENE, and margin status. The authors suggest that, although patients with high nodal burden (six or more positive nodes) benefit the most from concomitant RTCT, outcomes for these patients remain poor, even with multimodal therapy, so further adjuvant treatment intensification in this “very-high-risk” subset may optimize outcomes, representing an ideal population for testing of novel adjuvant therapeutic strategies, such as the combination of postoperative CRT with immune checkpoint inhibitors or targeted molecular therapies. Similarly, Feng and colleagues [80] found that patients with high lymph node ratios coupled with high numbers of positive lymph nodes who received POCRT had better 5-year DFS than patients who received surgery alone. The authors concluded that a high lymph node ratio is closely correlated with adverse parameters that markedly hinder prognosis. Fan et al. [82] explained the discordance of their results with the pooled analysis of EORTC/RTOG in different ways: first, the cancer type in their study was solely OSCC, whereas OSCCs made up only approximately 25% of cancers in the two classic randomized trials. Second, their study found that the benefit became significant after the total dose of cisplatin reached the dose of at least 200 mg/mq. The dose of chemotherapy was reduced in 20% of the patients in the POCRT arm of the two randomized trials. Third, most of the patients in their study had other RFs for tumor recurrence. Trifiletti et al. [77] found that the presence of multiple positive nodes seems to be a significant adverse prognostic factor in this setting and may be an appropriate selection factor for future prospective studies of adjuvant therapy in high-risk resected HNC to evaluate if nodal burden should be classified as a major risk factor rather than intermediate. In the study by Lin et al. [85], CGMH guidelines suggested POCRT for cancers with the presence of pN2, considering the presence of multiple pathologic nodes as already being a major RF. The only prospective trial that we included in our analysis [75] effectively showed a 2-year OS of 88% after POCRT in patients with histologic involvement of two or more lymph nodes.

#### 4.1.6. pT3–pT4 Tumors

Compared with the previous American Joint Committee on Cancer (AJCC 2010, 7th edition) staging system for oral cavity cancer, the most recent AJCC (2017, 8th edition) introduced a T-status reclassification [115,116].

In particular, the presence of a depth of invasion (DOI) > 5 mm but ≤10 mm results in a tumor upstaged T2 classification, whereas a DOI > 10 mm leads to an upstaged T3 classification. Effectively, the prognosis of cancers previously classified as pT2 can be remarkably worse if the DOI is high. Kano et al. reported 10-year CSS of 62.9% vs. 86.0% using a DOI cut-off of 10 mm [117]. Jung et al. also demonstrated a similar trend in 5-year OS (60% vs. 93%) and CSS (59% vs. 82%) when stratified by DOI of 9 mm [118]. In the same way, Newman and colleagues showed that DOI is a predictor of poor survival outcomes in pT3N0M0 OCC patients: OS and CSS are significantly worse in pT3N0M0 patients with deep DOIs (10–20 mm) [53].

The AJCC 2017 staging system also recommends that extrinsic muscle invasion (EMI) should not be considered as indicating T4 tumors but that this classification relates to tumors > 4 cm with DOI > 10 mm or invading the cortical bone of the mandible or maxilla, the maxillary sinus or the skin of the face. Stage IVA of OCC is characterized as a heterogeneous group of tumors, ranging from pT1-2 N2 tumors to pT4aN0 and pT4aN1-2 tumors. Using reconstructive surgery, it is nowadays possible to achieve adequate surgical margins regardless of tumor size. Liao et al. [119] clearly noted that survival rates for pT4N0 patients are better than pT4N+ tumors and, like those of patients with stage III, found similar 5-year LC and neck control rates, DFS, and OS rates between pT4N0 and pT1-3 N1 patients. The authors emphasized that adequate surgical margins should be obtained in pT4N0 patients to ensure a good prognosis, and we can thus argue that pT4 alone, similarly to pT3 alone, is not sufficient to nominate a patient for POCRT. Furthermore, this study confirms previous observations in the same group regarding the independent prognostic significance of lymph node metastases in OCC [120].

Previous studies by Liao et al. highlighted the idea that not all T4b tumors are equally amenable to surgical intervention. OCCs extending to the masticator space or pterygoid plate that were completely removed had comparable outcomes to T4a tumors [31,121]. The retrospective study by Patel et al. [83] showed using multivariate analysis that surgery plus POCRT was associated with outcomes superior to definitive RTCT, while surgery plus PORT was not. They concluded that the evidence for adding chemotherapy for very advanced tumors that are removed with negative margins is not clear, but one third of patients in their cohort (who had negative margins and no ECE) received POCRT, and the authors found that this approach appears to be associated with better survival and might justify further prospective trials to investigate the role of POCRT beyond the established major risk factors. Therefore, the pT4b stage could be a risk factor to consider for the addition of concomitant CT to PORT, considering the fact that confirming true negative margins in large tumors might be difficult and the possibility that unidentified microscopic disease (one of the two classical major risk factors) could exist. Certainly, with the presence of other minor risk factors in the pathological specimens, POCRT can be advised in this group of patients. These results are confirmed by the previous work by Spiotto and colleagues [76], which demonstrated an advantage in survival for pT3 and pT4 oral tongue cancers treated with POCRT. Like Patel, they supposed that advanced tumor classification may signify the likelihood of positive or very close margins, which may be difficult to identify in expansive tumors. In their work, Chen et al. [78] suggested that, in patients with more advanced tumors, such pT4b or pN3, but carrying no major RFs, POCRT must be considered in clinical practice.

#### 4.1.7. Low Neck Positive Nodes (Levels IV–V)

In the EORTC trial, only a quarter of all HNSCC patients were OCC patients with involved lymph nodes at levels IV–V. Additionally, the RTOG trial did not find a high risk of recurrence in these patients. Therefore, the benefit of POCRT in OCC patients with level IV–V node metastases is controversial.

It is generally assumed that level IV/V lymph-node metastasis in oral cancer occurs secondarily to level I–III node metastasis. Köhler et al. reported that multiple node metastases were a significant risk factor for lower neck metastasis [122]. Furthermore, it is anatomically possible for metastasis at level III or IV to flow directly from the anterior part of the tongue without first passing through the level I or II nodes. These cases may be associated with a skip metastasis, first reported by Byers et al. [123] for tongue cancer. Some investigators, such as Marchiano [124], have reported a poor prognosis in patients with metastasis at the lower neck level, and they suggest multimodal treatment for these patients. Jones [125] and Liao [126] showed that lower levels of lymph node involvement were associated with poor prognosis and higher risk of distant metastasis. In the study by Hasegawa and colleagues [19], multivariate analysis demonstrated significant relationships between level IV/V metastasis and prognosis in OCC patients. Tongue tumors, pN2 or pN3, and moderate or poor differentiation were significantly associated with the development of level IV/V metastases. The PORT and POCRT groups were associated with better 3-year cumulative DSS and OS rates than the surgery-only group. Adjuvant therapy (RT alone or POCRT) after surgery was therefore recommended by these authors for patients with level IV/V metastasis.

Thus, the indication for POCRT in the presence of level IV–V positive neck nodes could be relevant, but only if other minor RFs are present at the same time.

#### 4.1.8. Close Surgical Margins

Surgery for oral cancers is finalized to a satisfactory combination of complete tumor excision with adequate resection margins and preservation of healthy tissue involved in functional and cosmetic outcomes [127].

While the current definition of “clear surgical margin” is “a histopathologic margin of >5 mm”, a distance of tumor cells from surgical margins of 1–5 mm defines a “close surgical margin” [6]. Even if a close surgical margin could be considered a risk factor for tumor local recurrence due to potential residual microscopic disease in the surrounding tissue, this rationale is not associated with an evidential basis [128]. Similarly, recent data have suggested no impact from close surgical margins on survival outcomes [129].

Different reporting modalities for margin status among clinical trials represent a specific concern that could affect a clear understand of disease outcomes when comparing published studies. This consideration could produce subsequent controversies regarding the appropriate indication for postoperative approaches according to margin status [130]—in particular, limited consensus still exists on the appropriate RT fractionation regimen and radiation volumes [130].

Additional bias could affect the assessment of tumor borders and margins, including [131] their processing, and, eventually, lead to inadequate orientation of the tumor specimen—in many cases, aggravated by complex tumor shapes and sites. Furthermore, variable capacities in margin status prediction could result from the different intraoperative methods used to assess resection margins [127] and postresection shrinkage [131]. In particular, the reduction of tumor margin measurements (shrinkage), which occurs between the surgical pre-incision/excision processes and the histopathological evaluation [132], has been proposed as a potential cause for close pathological margins and requires prudential considerations by surgeons at the initial planning of resection margins.

Among six studies assessing the impact of margin status on patients’ outcomes that were evaluated in the present review [7,9,13,77,78,79], two analyses [9,78] suggested that a close margin (≤5 mm) is a significant poor prognostic factor for survival and recurrence (OS [9,78], DSS [78] and LRC, DMF, DFS, and OS [78]). A margin distance ≤4 mm was reported to be a significant poor prognostic factor (for local–regional recurrence) by Fan et al. [7]. Furthermore, Lin et al. [85], based on data from few patients (only four), speculated about the impact of close margins (≤4 mm) on patients’ outcomes.

According to these data, OSCC patients with this additional risk factor may benefit from postoperative RT/CCRT [9,78].

#### 4.1.9. Other Minor Risk Factors

To improve therapeutic chances and indications, the presence of additional minor risk factors (tumor differentiation grade, bone and skin invasion, tumor location, growth pattern/pattern of invasion) should be take into account.

Poor differentiated tumor grade has already been identified as an adverse prognostic factor (mainly for neck relapse) [85]. Lee et al. [14] confirmed that poor differentiated tumors are at increased risk of relapse at regional and distal sites and encouraged further analyses to support the hypothesis of a synergy between poor tumor differentiation and ENE in order to identify patients suitable for more aggressive postoperative approaches.

In a study by Chen [78], the adverse impact of poor disease differentiation on 5-year LRC, DMF, DFS, and OS was confirmed either in univariate or multivariate analyses. Feng and al. [80] reported a close correlation between pathological grade and 5-year DFS (HR: 1.238, 95%CI: 1.031–1.486; *p* = 0.022) and 5 year DSS (HR: 1.307, 95%CI: 1.073–1.593; *p* = 0.008). Data from Fan’s study [79] also suggested an inferior outcome after surgery and PORT in OSCC patients with poor differentiation as an additional risk factor.

Only a few of the included studies have analyzed and confirmed the prognostic role of bone invasion [7,79,82] and skin invasion [79,82]; in particular, in the univariate analysis by Fan et al. [82], those factors were correlated with poor OS (*p* < 0.05), RFS (*p* < 0.05), and LRRFS (*p* < 0.05).

A SEER Database analysis [133] aiming to determine if tumor subsite predicted survival confirmed that upper/lower gum, retromolar trigone, and floor of mouth subsites showed better survival rates compared to oral tongue. The authors reviewed and underlined a specific propensity for locoregional spread and a specific tendency for more advanced tumor staging (e.g., bone invasion) according to the anatomic location of the tumor and its subsite [133]. Among the studies included in the present review, the univariate analysis by Fan et al. [7] showed a significant correlation (*p* < 0.05) between tumor location (at the hard palate and retromolar trigone) and local–regional recurrence.

Further considerations on cancer-specific responses to therapies or tissue radioresistance could be useful to clarify the prognostic impact of tumor location and to improve tailored treatments.

Limited data on the predictive and prognostic role of tumor growth pattern/pattern of invasion are available. Tumor growth pattern (which can be defined as exophytic, ulcerative, or infiltrative [80]) was reported to be closely correlated with 5-year DSS (HR: 1.244, 95%CI: 1.067–1.450l; *p* = 0.005) in the univariate analysis by Feng et al. [80]. In the study by Yamada et al. [13]—besides the potential bias due to the variability in the pathologists’ determinations and reporting—a decrease in survival rate was confirmed as the pattern of invasion progressed from grade 1 to 4 according to the Yamamoto–Kohama classification [134] (this is a score on histological findings obtained at the border between a tumor mass and normal tissue [13]).

### 4.2. Main Issues and Limitations

From the literature evidence and our analysis, it seems that a combination of various intermediate-risk features may increase the probability of failure for head and neck sites that are already more prone to recur, such as OCC. Practice guidelines [6] suggest that either POCRT or PORT may be appropriate for patients with other minor risk factors.

The results of this meta-analysis of data collected using a systematic review of the literature show that adding concomitant chemotherapy to adjuvant radiotherapy in patients with resected OCC, who have no ECE and no positive margins but only minor risk factors, significantly improves OS. In our analysis, the DFS and LRFS of patients treated with POCRT were not statistically significant compared to patients treated with PORT alone; these results can be explained, especially considering that the number of the articles calculating the DFS or the LRFS was less than half of the studies globally included in the meta-analysis and, therefore, included a significantly lower number of patients, which reduced their statistical significance.

In fact, the LRFS of the POCRT group seems to be worse compared to the group treated with PORT alone: this trend could be explained by the fact that patients treated with POCRT could have multiple minor RFs that worsen their prognosis despite the addition of concomitant CT.

However, the significant increase in the OS of patients treated with POCRT appears to be relevant because, to date and to our knowledge, this is the only meta-analysis that has analyzed oncological outcomes in resected OCC patients with solely minor pathological RFs. Favorable data about the OS in a “grey area” left by the results of the classical EORTC/RTOG prospective studies about POCRT force clinicians involved in multidisciplinary decisions to carefully consider the addition of concomitant chemotherapy to postoperative radiation therapy in patients with only minor RFs.

Some of the selected papers [78,79,85] including only OCC patients suggest that the risk of disease recurrence in OCC with solely minor RFs does not depend on a single RF but on the coexistence of multiple minor RFs, as also shown in a previous work published by Fan and colleagues [7]. The coexistence of multiple special pathologic characteristics indicates a poor prognosis, even if these characteristics are not significant when analyzed separately. The presence of a greater number of risk factors correlates with an increased risk of tumor recurrence in various retrospective analyses [135]; thus, this risk may be cumulative. In their work published in 2017, Fan et al. [79] concluded that POCRT increases recurrence-free survival and OS in patients harboring three or more minor RFs. Subgroup analysis performed in the study by Chen et al. [81] showed that patients with at least three minor RFs or at least one major RF who received POCRT had significantly better LCR, DFS, and OS compared to patients receiving PORT alone. The study by Lin et al. [85] showed that therapeutic decisions in which the adjuvant combined treatment was performed in the presence of positive margins, ECE, pN2, or at least three minor RFs calculated on the basis of a cumulative scoring system made it possible to better individuate the patients that benefited from POCRT in terms of oncological outcomes. This indicates the need for the introduction of cumulative scores in the guidelines to suggest more precisely to clinicians whether the opportunity to add a concomitant CT to the adjuvant RT in cases of resected OCC with solely minor RFs exists or not. Obviously, any decision about concomitant chemotherapy must be confirmed considering other factors, such as performance status (PS), existing comorbidities, and the biological age of the patient. In their work, Chen et al. [81] concluded that POCRT may offer a survival benefit for non-oropharyngeal intermediate-risk advanced-stage head and neck patients < 70 years of age with pT1-4N2-3 disease but may not benefit those ≥70 years of age. As POCRT has been reported to have 33% greater toxicity (mainly grade 3 acute) than only PORT [136], the selection of patients to address with an intensified approach is crucial.

Several issues that arise when considering the role of chemotherapy added to PORT are the timing, the variety of drugs, and the schedules used. Since the National Cancer Database (NCDB) does not provide information on chemotherapy, the benefits observed in OS by Trifiletti [77] and Spiotto [76] could reflect the absence of data on chemoradiotherapy toxicity.

In Fan’s study (2014) [82], a total of six patients (2.2%) died due to acute or late toxicity but as reported, POCRT did not correlate with lethal toxicity, neither during PORT (Fisher’s exact test, *p* = 0.587, one-sided) nor across all time (Fisher’s exact test, *p* = 0.237, one-sided). Fan (2017) [79] reported 2.8% mortality in the POCRT group. These results are similar to those of randomized clinical trials (EORTC/RTOG). In his study, Fan [82] showed that POCRT is likely to reduce the risk of recurrence, mainly with a cumulative cisplatin dose of 200 mg/m^2^. If we note that, in the two randomized clinical trials (EORTC/RTOG), the dose of chemotherapy was reduced in 20% of population, then it is possible that this could have impacted the results. However, the correlation between the compliance of chemotherapy and treatment outcomes was not analyzed since those trials were not designed to answer such questions.

The retrospective nature of almost all the studies included in the meta-analysis limited our results. In the selected studies, data about the weight of global PS, concomitant comorbidities and acute and late toxicities were lacking, thus restricting the analysis on the oncological benefit of POCRT. Finally, the use of an intensified approach (POCRT) in the series analyzed might suggest a selection bias favoring the results of the POCRT group with less comorbidities and better PS. However, as shown in Table 3, in half of the studies there were comparable Charlson scores/comorbidities or ECOG in the two groups of patients (PORT vs. POCRT). Likewise, it could be said that an intensified approach was pursued for patients deemed to have a greater risk of recurrence of disease in relation to risk factors. Trifiletti [77] notes that almost half of patients in the NCDB were subjected to POCRT in the absence of major risk factors, thus mirroring a precautionary approach to HNSCC patients with minor risk factors. Furthermore, the worldwide introduction of intensity-modulated RT has reduced the toxicities of combined treatments (with respect to the previously adopted 3DCRT) [137] by improving the dose-sparing of critical normal tissues; this advantage has to be taken into account in clinical practice, despite the lack of data reporting RT techniques in the analyzed studies. Prospective data collected from larger patients’ cohorts could improve current knowledge and support therapeutic management of OCC patients with specific/multiple minor risk factors.

## 5. Conclusions

Our meta-analysis shows a statistically significant (*p* < 0.0001) improvement in the OS of patients with resected oral cavity carcinoma treated with postoperative chemoradiotherapy in the presence of solely minor pathological risk factors. Future and ongoing prospective studies could confirm the role of concomitant chemotherapy with postoperative radiotherapy in this patient setting, not only based on patient-related factors (PS, comorbidities, age) but also cumulative scores considering different minor risk factors.

## Figures and Tables

**Figure 1 cancers-14-03704-f001:**
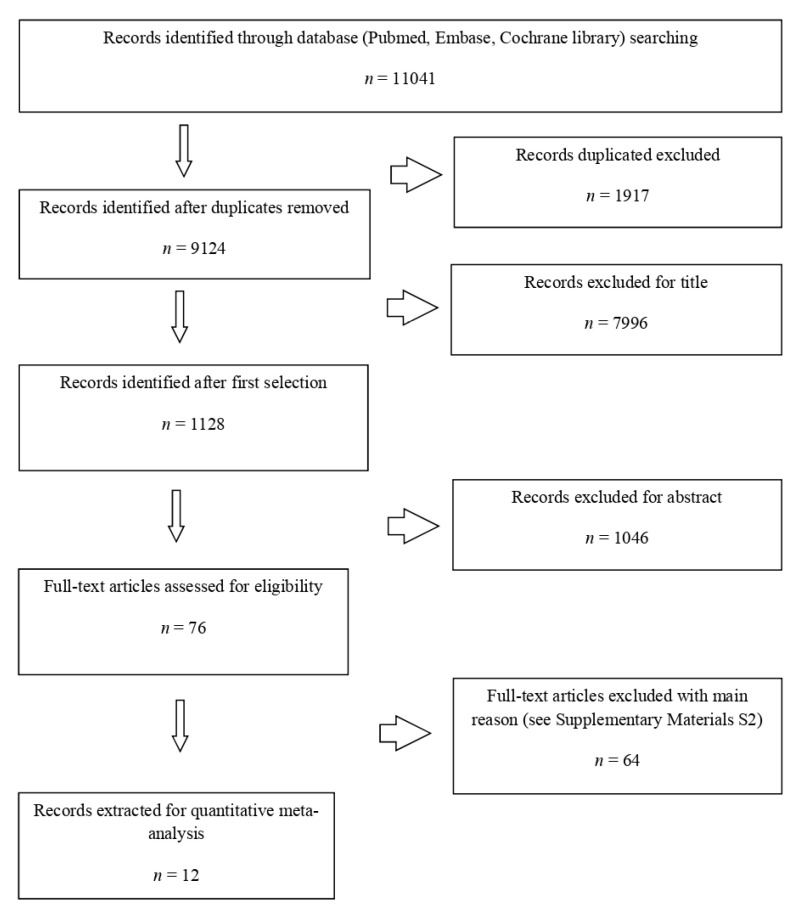
PRISMA workflow for the systematic review.

**Figure 2 cancers-14-03704-f002:**
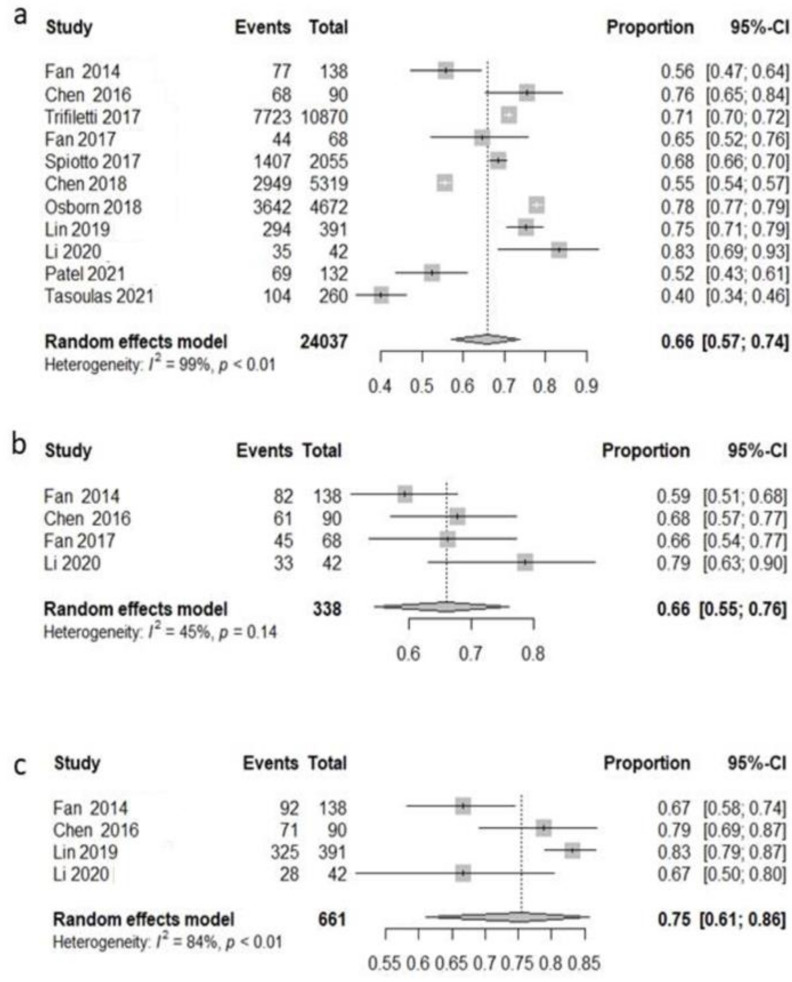
Forest plot, 3-year survival analysis, all patients included. Overall survival (OS) (**a**); disease-free survival (DFS) (**b**); local recurrence-free survival (LRFS) (**c**).

**Figure 3 cancers-14-03704-f003:**
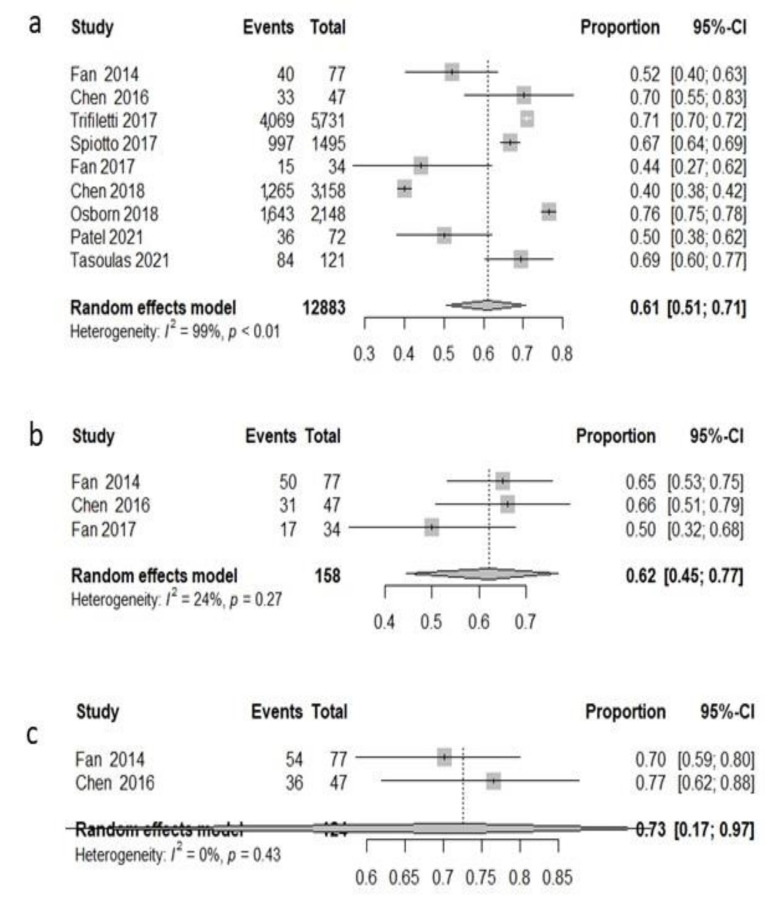
Forest plot, 3-year survival analysis, only RT population: overall survival (OS) (**a**); disease-free survival (DFS) (**b**); local recurrence-free survival (LRFS) (**c**).

**Figure 4 cancers-14-03704-f004:**
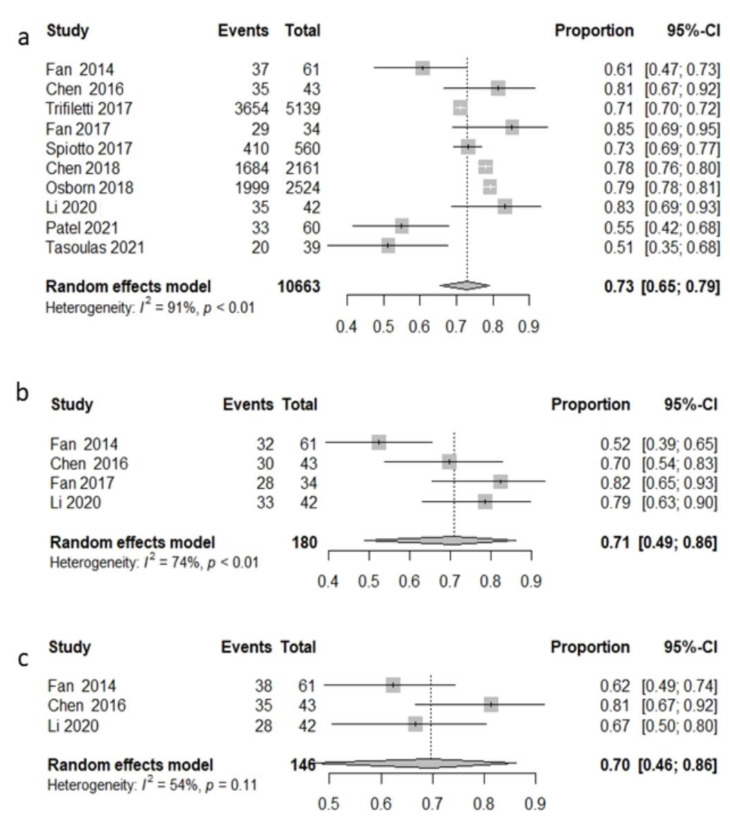
Forest plot, 3-year survival analysis, chemoradiotherapy population: overall survival (OS) (**a**); disease-free survival (DFS) (**b**); local recurrence-free survival (LRFS) (**c**).

**Figure 5 cancers-14-03704-f005:**
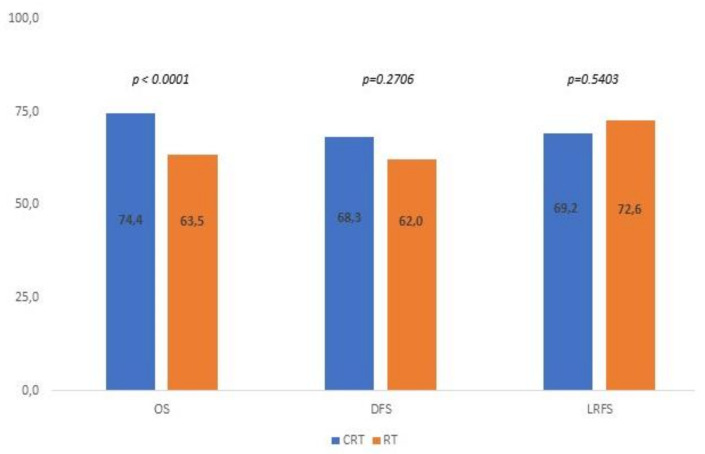
*p* values for OS, DFS, and LRFS for all analyzed studies.

**Table 1 cancers-14-03704-t001:** Criteria for study selection according to PICOT model.

Selection Criteria	Inclusion Criteria	Exclusion Criteria
P Population	Adults (age > 18 years) with resected non-metastatic squamous OCC	Pediatric patients (age < 18) and histology other than SCC
I Intervention	Postoperative radiotherapy alone (PORT)	Post-operative chemo-radiotherapy (POCRT)
C Comparison	Squamous OCC with minor/intermediate pathological risk factors	Squamous OCC with major pathological risk factors (positive margins and/or ECE)
O Outcome	OS, DFS, LRFS	
T Timing	2000–2021	

**Table 2 cancers-14-03704-t002:** Characteristics of studies included in the meta-analysis.

Author, Year	Oral Cavity Subsite	Number of Patients	Accrual Period	Types of Minor Pathological Risk Factors	Presence of Major Pathological Risk Factor	Number of Patients in CRT Arm/Total	Median FU (Months)	Disease-Free Survival (DFS)	Local Recurrence-Free Survival (LRFS)	Overall Survival (OS)
Spiotto M. (2017) [76]	Oral tongue	2803	2004–2012	LVI, DOI ≥ 5 mm, pT3 or pT4, multiple lymph nodes without ENE	No	1308/2803	33	/	/	73.3% HR: 0.78 (95%CI: 0.64–0.96)
Trifiletti (2017) [77]	Oral cavity and other H and N sites (oropharynx, larynx, etc.)	5094/10870 (2899 RT, 2195 CTRT)	2004–2012	Positive node at level IV or V, multiple lymph nodes without ENE	No	2195/10870	38,4	/	/	3 y OS: 74.2% 5 y OS: 65.3% HR: 0.902 (95%CI: 0.861 to 0.944)
Chen W.C. (2016) [78]	OSCC	567	2002–2013	PNI, LVI, DOI ≥ 5 mm (10 mm), close margin (< 2–5 mm), pT3 or pT4, multiple lymph nodes without ENE	1 (positive margins in 28 patients, ENE 83 patients)	127/567	42	50.2% HR: 0.38 (95%CI: 0.21–0.68)	74.5% HR: 0.33 (95%CI: 0.14–0.78)	59.8% HR: 0.37 (95%CI: 0.19–0.72)
Fan K.H. (2017) [79]	Buccal mucosa, tongue, gums, retromolar trigon, mouth floor, hard palate	68 of 109 initially selected (34 CRT, 34 RT)	1999–2009	PNI, LVI, DOI ≥ 5 mm, close margin (<2–5 mm), pT3 or pT4	No	34/68	86.4	75.4%	75.4% HR: 0.248 (95%CI: 0.103–0.596)	67.2% HR: 0.426 (95%CI: 0.212–0.858)
Feng (2017) [80]	Tongue, gingiva, buccal mucosa, mouth floor, hard palate	809 (14% oropharynx)	/	PNI, LVI, pT3, or pT4, multiple lymph nodes without ENE	Yes, ENE +	114/809	Not reported	51.4%	/	/
Chen M.M. (2018) [81]	Lips, oral cavity	5319 total H and N patients. Oral cavity: pRT group 1571, pCRT 956	2010–2013	LVI, pT3, or pT4, multiple lymph nodes without ENE	No	956/1571	Not reported	/	/	For T1–4 N2–3, HR: 0.73 (95%CI: 0.58–0.93). For T3–4 N0–1, HR: 0.92 (95%CI: 0.71–1.19)
Fan K.H. (2014) [82]	Tongue, buccal mucosa, gums, retromolar trigone, mouth floor, hard palate, lips	138	1998–2008	PNI, LVI, DOI ≥ 5 mm, close margin (<2–5 mm), positive nodes level IV or V	No	77/138	35	60%	70%	60%
Li R. (2020) [75]	Tongue, gingiva, buccal mucosa, mouth floor, retromolar trigone, palate, lip	91	2016–2018	pT3 or pT4, multiple lymph nodes without ENE	Yes, positive margins and ENE +	91/91	24	75.3% (95%CI: 65.7–84.2%)	79.0%	82.4% (95%CI, 73.0–89.6%)
Patel (2021) [83]	Retromolar trigone, gum, cheek mucosa, mouth floor and NOS, tongue, vestibule, lip	1338	2004–2017	pT3 or pT4	Yes, positive margins and ENE +	163/1338 (the other 23 patients received neoadjuvant CT then surgery +RT)	24	/	/	64.6%
Osborn (2018) [84]	OCSCC	2303	2004–2012	pT3 or pT4	Yes, positive margins and ENE +	1381/2303	47,7	/	/	67.4%
Lin C (2019) [85]	OCSCC	1200	2004–2016	pT4, DOI > 5 mm (5 mm), positive nodes level IV or V, PNI, LVI	Yes, positive margins and ENE +	411/1200	61	75%	/	83%
Tasoulas (2021) [86]	OCSCC	616, 167 for OC	2002–2006	LVI, PNI, T3 or T4, multiple lymph nodes without ENE	Yes, ENE +	92/616 45 high-risk patients	Not reported	/	/	HR: 0.30 (95%CI: 0.15–0.61) for high-risk patients

**Table 3 cancers-14-03704-t003:** Weights of comorbidities in all the included studies and percentages of each minor risk factor in studies with only OCC population.

Study	Comorbidities	PNI	LVI	Multiple Nodes	pT3–T4	DOI	Close Margins	pN1	Low Neck Nodes
Fan 2014 [82]	NE	50% POCRT 25% PORT	17% POCRT 8% PORT	100% POCRT 100% PORT	53% POCRT 26% PORT	≥10 mm in 65% POCRT 67% PORT	32% POCRT 23% PORT	No patients	4% POCRT 5% PORT
Spiotto 2017 [76]	Charlson index Well-balanced in POCRT vs. PORT	NE	10% POCRT 9% PORT	56% POCRT 36% PORT	25% POCRT 18% PORT	≥5 mm in 4% POCRT vs. 8% PORT	NE	30% POCRT 58% PORT	NE
Fan 2017 [79]	ECOG 0–1 in 97% POCRT 94% PORT	62% POCRT 65% PORT	15% POCRT 6% PORT	No patients	pT4: 62% POCRT, 65% PORT	≥10 mm in 75% POCRT 94% PORT	56% POCRT 74% PORT	53% POCRT 26% PORT	NE
Chen WC 2016 (subgroup analysis) [78]	NE	NPE for patients with only minor RFs	NPE for patients with only minor RFs	NPE for patients with only minor RFs	NPE for patients with only minor RFs	NPE for patients with only minor RFs	NPE for patients with only minor RFs	NPE for patients with only minor RFs	NPE for patients with only minor RFs
Patel 2021 [83]	No comorbidity in 78% of patients	NE	NE	53% of patients	100% pT4b	NE	NE	11% of patients	NE
Li 2020 [75]	Not specified in the study	NPE for patients with only pN2	NPE for patients with only pN2	NPE for patients with only pN2	NPE for patients with only pN2	NPE for patients with only pN2	NPE for patients with only pN2	NPE for patients with only pN2	NPE for patients with only pN2
Lin 2019 [85]	NE	49% patients with minor RFs	6% patients with minor RFs	13% patients with minor RF	28% pT3 44% pT4 patients with minor RFs	≥10 mm, 64% patients with minor RFs	13% patients with minor RFs	18% patients with minor RFs	0.7%patients with minor RF
Trifiletti 2017 [77]	Charlson Index: 0: 78% PORT, 82%POCRT; 1: 17%PORT, 15% POCRT; 2: 4% PORT, 3% POCRT	NPE for studies with mixed populations of HN cancers	NPE for studies with mixed populations of HN cancers	NPE for studies with mixed populations of HN cancers	NPE for studies with mixed populations of HN cancers	NPE for studies with mixed populations of HN cancers	NPE for studies with mixed populations of HN cancers	NPE for studies with mixed populations of HN cancers	NPE for studies with mixed populations of HN cancers
Tasoulas 2021 [86]	NE	NPE for studies with mixed populations of HN cancers	NPE for studies with mixed popultions of HN cancers	NPE for studies with mixed popu- lations of HN cancers	NPE for studies with mixed populations of HN cancers	NPE for studies with mixed populations of HN cancers	NPE for studies with mixed populations of HN cancers	NPE for studies with mixed populations of HN cancers	NPE for studies with mixed populations of HN cancers
Feng 2017 [80]	NE	NPE for studies with mixed populations of HN cancers	NPE for studies with mixed populations of HN cancers	NPE for studies with mixed populations of HN cancers	NPE for studies with mixed populations of HN cancers	NPE for studies with mixed populations of HN cancers	NPE for studies with mixed populations of HN cancers	NPE for studies with mixed populations of HN cancers	NPE for studies with mixed populations of HN cancers
Osborn 2018 [84]	Charlson Index: 0: 79% PORT, 81% POCRT; 1: 16% PORT, 15% POCRT; 2: 5% PORT, 3% POCRT	NPE for studies with mixed populations of HN cancers	NPE for studies with mixed populations of HN cancers	NPE for studies with mixed populations of HN cancers	NPE for studies with mixed populations of HN cancers	NPE for studies with mixed populations of HN cancers	NPE for studies with mixed populations of HN cancers	NPE for studies with mixed populations of HN cancers	NPE for studies with mixed populations of HN cancers
Chen MM 2018 [81]	Comorbidities: 0: 37% PORT, 39% POCRT; 1: 10% PORT, 9% POCRT; 2 or +: 2.5% PORT, 2% POCRT	NPE for studies with mixed populations of HN cancers	NPE for studies with mixed populations of HN cancers	NPE for studies with mixed populations of HN cancers	NPE for studies with mixed populations of HN cancers	NPE for studies with mixed populations of HN cancers	NPE for studies with mixed populations of HN cancers	NPE for studies with mixed populations of HN cancers	NPE for studies with mixed populations of HN cancers

Legend: NE: not evaluated; NPE: Not possible to extrapolate; RF: Risk factor; HN: head and neck.

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
