# Peer review of "Adding Concomitant Chemotherapy to Postoperative Radiotherapy in Oral Cavity Carcinoma with Minor Risk Factors: Systematic Review of the Literature and Meta-Analysis"

_cancers, 2022, doi:10.3390/cancers14153704_

Round 1

Reviewer 1 Report

The authors conducted a systematic review and meta-analysis of concomitant chemotherapy with postoperative radiotherapy in oral cavity carcinoma.

It is unclear what “oral cavity carcinoma” means. Does it include adenocarcinoma derived from the minor salivary gland? More detailed explanation is needed.

The authors stated that they were reviewing CRT studies for oral cavity carcinoma with minor risk factors, but this manuscript appears to be an original-style article, not a review article. For review article, the style should be changed to match the journal. At least, the subtitles, Materials and methods and Results, should be deleted, and the authors should re-write the text.

Patients were treated in these studies from 2004-2018. During this period, there was the opportunity to introduce IMRT worldwide (page 190, ref. 12, 13). It is necessary to discuss how this change in RT affected the efficiency and side effects of RT.

In the case of original paper, the authors should clearly describe what is the novel point of this study.

For the original paper, discussion: sections 4.1.1-4.1.9 is too long. Over-explanation of each minor risk factor should be minimized. This part did not explain the results of this study. The volume of this part should be reduced by about half. 

The actual discussion is focused on Section 4.2. Most importantly, the additional effect of chemotherapy on RT. The discussion of the final result (Figure 5) is insufficient. There are no significant differences between the groups, but with LRFS, patients treated with RT only showed a better prognosis compared to the CRT group. However, with DFS and OS, the results were reversed. The authors need to explain what these results mean. RT can have a significant impact on residual tumor cells at the surgical margin. However, its effect may not be enough to control dormant lymph node metastasis. This kind of discussion is required.

Analysis of LRFS in patients treated with RT alone is performed using only two studies (Figure 2). In addition, Fan-KH et al. published two papers in 2014 and 2017. The authors need to explain the difference between these two studies.

Page2, lines 68 and 73: There is a duplicate of (ECE). When it first appears, the authors need to spell out.

Page 20, line 595: What does intermediate in minor/intermediate pathological RF?

Reviewer 2 Report

The aim of the submitted systematic review with meta-analysis is to assess the potential tole of chemotherapy added to post-operative radiotherapy in locally advanced oral cavity carcinoma with minor risk factors (perineural invasion, lymph vascular invasion, pN1 single, depth of invasion more than 5 mm, close margin les than 2 - 5 mm, node positive IV or V level, pT3 or pT4, multiple lymph nodes in absence of extranodal extension). 

The topic is very important, because benefit of addition of concomitant chemotherapy to the postoperative radiation therapy in oral cancer with only minor risk factors is not clear and the decision is mostly subjective. 

PRISMA methodology was used for the review and adequate methods were used for meta-analysis. 

Meta-analysis showed a statistically significant improvement in the OS of patients with resected oral cavity carcinoma, treated with postoperative chemoradiotherapy in presence of solely minor pathological risk factors. 

The authors suggest need for the introduction of cumulative scores considering different minor risk factors in the guidelines nad perspective studies on this topic. 

The discussion is comprehensice and conclusions are clear. 

I have minor comments: 

Page 5 last paragraph - "The analyzed population of each study varied greatly, ranging from 425 to 108706 patients" - the number 108706 seems not to be correct. 

Page 14, lines 282 and 283 - "In a multi-institutional collaborative group analysis (196 POCRT, 128 PNI) /38/. On multivariate analysis, DFS...." - syntax seems not to be correct. 

In general, the article is interesting, with new informations, well writen. I recommend to accept. 

Round 2

Reviewer 1 Report

The authors respond appropriately to the points raised by the reviewer.